# Resilient Urbanization: A Systematic Review on Urban Discourse in Pakistan

**Latif Abdul * and Tao-fang Yu**

Department of Urban Planning, School of Architecture, Tsinghua University, Beijing 100084, China;
yutaofang@tsinghua.edu.cn

\* Correspondence: marilatif@yahoo.com

**Abstract:** Urbanization is a common phenomenon in the modern world. It has come with new challenges, especially for developing countries. Such countries, therefore, have to stay ahead in their preparedness efforts to meet these urban issues halfway. Unfortunately, urban residents in Pakistan are living in serious social, physical, and economic hardships. Despite being economic engines, cities in Pakistan suffer from stresses like climate change, haphazard and unregulated expansion, housing shortage, and a lack of basic civic amenities. While using systematic review methodology, we collected published and grey data from national and international sources. Literature shows that successive governments in Pakistan gave ample space to urban development in most of the policy documents. However, urban resilience and community engagement were given scant attention. This major gap, both in policy and practice, needs to be bridged to promote resilient and sustainable urbanization in Pakistan.

**Keywords:** Pakistan; sustainability; urbanization; urban resilience; urban plans; urban challenges

---

## 1. Introduction

"At no time in human history have so many people lived in cities. Poor land-use planning; environmental management will increase risk and exacerbate the effects of natural disasters." Kofi Annan.

Urbanization is a complex socio-economic process that transforms the built environment, converting formerly rural areas into urban settlements, while also shifting the spatial distribution of a population from rural to urban areas. The major consequences of urbanization are increasing population size or urban settlements, and in the number and share of urban residents compared to rural dwellers [1,2].

Urban resilience is the capacity of cities to act efficiently so that its residents and workforce, especially the vulnerable people, survive and thrive in spite of the stresses or shocks they encounter in their everyday lives [3,4]. Similarly, 100 resilient cities, a not-for-profit organization of the Rockefeller Foundation, defines the term resilient urbanism as surviving and thriving, regardless of the challenge. The word resilience means "the persistence of relationships within a system" and "the ability of these systems to absorb the changes of state variables, driving variables, and parameters, as well as persist [5,6].

Two distinctly evident but intertwined trends of the 21st century are rapid urbanization and frequent natural disasters [7]. A densely populated and interconnected world requires new models of governance to manage rapid urban growth and mitigate stresses such as extreme weather events, the refugee crisis, disease pandemics, and cyber-attacks [8]. Acute or sudden shocks such as earthquakes, hurricanes, and terrorist attacks are further exacerbated by chronic stresses such as recurrent flooding, high unemployment, and overtaxed or inefficient public transportation [8]. With the occurrence of any



climatic activity, urban economic centers will bear serious consequences [9]. Therefore, current and future development initiatives must prioritize resilient urban development.

The resilience of the urban population, the magnitude of disasters, available support, and degree of preparedness define the impact of a natural disaster. An adverse event will not turn into a disaster if it strikes a resilient population [10]. On the other hand, a vulnerable residential area can bear seriously disastrous consequences even in developed countries [11]. Thus, it depends on the system's resilience whether it can bear stresses or face serious physical and economic losses in the event of climatic or sociopolitical stress.

Indeed, disasters will keep coming with increasing frequency and ferocity. Tsunamis in Japan and Sumatra (an Indonesian island), heavy floods in Pakistan, and raging forest fires in Australia, which took tens of thousands of lives, are only some of the many tragic catastrophic events of the recent past. The issue is not climate change alone. There are many other global challenges, which interplay with interdependent and rapidly globalizing human societies [6]. Thus, climate change mingles with other global challenges to bring disasters.

The importance of cities cannot be overemphasized as urbanization and per capita productivity and therefore income have a positive correlation [12]. The higher the GDP per capita income of a country, the greater the rate of urbanization. For instance, rich countries like Belgium, Switzerland, Luxemburg, and The Netherlands are comparatively more urbanized than developing ones like Moldova, Serbia, Bosnia-Herzegovina, and Albania. Therefore, higher urbanization is directly proportional to national development [13].

However, urbanization has some serious downsides as well. Especially in developing countries, it has negative impacts like the degradation of natural habitats, disruption of hydrological systems, modification of energy flow, and nutrient cycling [14]. Hence, development must be sustainable so that it does not negatively affect the ability of future generations to live their lives. For the planet and cities to sustain the lives of *Homo sapiens,* people must cooperate at a scale unprecedented in human history [15].

Pakistan, a South Asian country, has one of the highest populations and urbanization growth rates in the world. Globally, it is the sixth most populous country [16], while economically it is 22nd in the world. Nevertheless, Germanwatch has declared the country as the fifth most vulnerable due to climate change despite emitting fewer greenhouse gases [17].

Cities of Pakistan have badly suffered from climatic catastrophes. For instance, Karachi, the largest city in the country, has faced the worst floods in August 2020, while summer heat in the year 2015 also killed hundreds of residents. Lahore, another megapolis of Pakistan, has attained the title of the third most polluted city in the world only behind Indian capital Delhi and Uzbek capital Tashkent [18]. Other Pakistani cities do not fare any better as they have their share of urban woes. Thus, barring a few exceptions like the capital Islamabad, most of the Pakistani urban areas are the least resilient to disasters. In the case of a minor calamity, they might suffer serious physical and economic losses.

In spite of serious urban challenges, little research has been carried out on resilient urbanization in Pakistan. Hence, a major research gap persists in the country's urban literature. This paper has attempted to bridge this gap while simultaneously highlighting a significant issue of resilient urbanization in Pakistan for future urban researchers.

The goals of this review paper were (1) to understand urbanization trends and urban resilience challenges in Pakistan, (2) undertake an appraisal of national urban literature, and (3) make a comparative analysis of the national urban literature and global urban resilience models (4) for suggesting policy recommendations.

## 2. Methodology

We used the systematic review method for this study. It is a means for evaluating and interpreting existing literature on a specific research question, area of interest, or topic so that a trustworthy and

rigorous secondary review is conducted [19]. The technique is more appropriate for an exhaustive topic like resilient urbanization and urban policy discourse in Pakistan.

Both published and grey literature was searched and included in the study for analysis. Published literature including national and international regulations, guidelines, models, and laws on urban resilience was considered. All plans and policies, regardless of their origin or impact, specific or general, urban-oriented or rural–urban mix, and their relevance to cities were studied and incorporated. Grey literature like unpublished reports, dissertations, and abstracts were carefully considered to minimize the research bias and maximize validity [20,21].

A study protocol was developed before starting the review. The protocol contained the inclusion/exclusion criteria of data to be analyzed. It also had a list of databases, the methodology of data extraction, and summary writing.

Inclusion and exclusion criteria were mainly based on research objectives. We used separate procedures for extracting data from national and international urban resilience and planning discourse. For the former category, we studied all national-level documents. These included Vision 2025, Framework for Economic Growth 2011, National Climate Change Policy 2012, Constitution of Pakistan for Legislation on Local Government, Census 2017 data, Urban Unit Lahore, and Orangi Pilot Project. For the latter category, only those urban resilience models and protocols were included which have been ratified or voluntarily adopted by Pakistan. Such adopted and ratified documents include but are not limited to urban data from the United Nations, Asian Development Bank (ADB), United Nations Integrated Strategy for Disaster Reduction (UNISDR), Munich Re, research journals on resilient urbanization, Sendai Framework, Paris Agreement, and Sustainable Development Goals (SDGs). The rest of the global urban resilience models and precedents were included using the key terms on the topic. It is followed by selecting the appropriate models using the snowball sampling method considering their context and applicability in Pakistan.

The key terms for research included "Pakistan", "sustainability", "disaster planning", "disaster preparedness", "urbanization", "resilient urbanization" "urban policy", "land use planning", and "resilience". The terms were identified using the research topic, research objectives, and reviews of the research results.

The selected literature was extensively perused for extracting relevant data. For instance, national development discourse contains the development plans of several sectors. However, we analyzed only those sections which contained terms like urbanization, urban planning, or resilience building. The rest of the chapters and sections were excluded. Likewise, data from global models were also extracted using the same technique. Extracted data were synthesized descriptively for drawing comparison and suggesting policy recommendations.

A comparative study has been conducted between national urban discourse with that of international urban resilience models and frameworks. It is followed by recommendations for the national urban policymakers.

## 3. Urban Discourse in Pakistan

### 3.1. Urbanization in Pakistan

With 210 million people today, Pakistan will be the fourth most populous nation in the world by 2030. In the 1998 census, the urban share of the total population in Pakistan was 32%, which is expected to be over 50% by 2025 under the administrative definition. However, Reza Ali, an eminent Pakistani urban scholar, while using satellite imaging, claims that 70% of Pakistan's population is non-rural [22]. He clarified that it does not mean that 70% is urban; rather this percentage of the population is living in concentrated areas in or around some urban core.

Research reveals that Pakistan will achieve rural and urban population parity around 2050. However, the majority of Pakistan's population is already living in real urban areas or highly dense agglomerations, which is factually not rural. In any case, Pakistan is predominantly urban.

Widespread floods across Pakistan in 2010 and 2011 forced the permanent migration of farmers from rural to urban areas of Pakistan [23], thus, further accelerating urbanization in the country.

However, rapid growth in cities has made them unable to absorb, comfortably accommodate, and meaningfully employ rural-to-urban migrants. It has therefore exacerbated the social/ethnic tensions in the cities. The immense challenge for Pakistan is to resolve the existing urban problems and plan for reaping the full economic potential of urbanization [24].

*3.2. Urban Challenges in Pakistan*

Pakistan faces a challenging urban environment. For example, Karachi, one of the top ten largest cities of the world with around 20 million population, confronts frequent power failures, water shortages, transport woes, heat island effect, rising sea levels, ever-expanding unregulated informal settlements, urban flooding, choked drains, and an extremely poor solid waste management system. Similarly, Lahore has recently witnessed the worst smog in recent history. The issue has given the city an unwanted distinction of the third most polluted city in the world, only behind its next-door neighbor Delhi and Central Asian Uzbek Capital Tashkent [18]. Karachi and Lahore are the two largest and most well-funded urban municipalities in Pakistan. The urban environment in other cities, except the federal capital Islamabad, is even worse than one could imagine.

In spite of these serious challenges and the recurrence of disasters, there is no disaster monitoring and early warning system in Karachi. Human and financial resource constraints, limited institutional capacities and coordination, and non-existing emergency operation centers have further deteriorated the city's fragile urban sustainability. A senior official of the Sindh government shared these thoughts at an international forum in September 2018 [25].

From the collected data, we found the following major urban challenges in Pakistan:

(a) Climate change: climate change is not only a big threat to urban resilience in Pakistan but it also affects almost the entire world. It has both direct and indirect effects. Direct ones include storms, typhoons, and heatwaves, the inundation of coastal areas due to sea-level rise, temperature increase, and disturbances in rainfall patterns. Indirect ones in urban areas include severe flooding resulting in road blockages, blackouts, risk of water or vector-borne diseases due to the accumulation of rainwater in the streets and low-lying areas, and rise in temperature which ultimately leads to health and infrastructure losses.

Damages from climate change have resulted in US$1.7 trillion global losses from 2000 to 2012 [3]. Similarly, EM-DAT (emergency disaster database), reports that the total number of natural disasters year-on-year has increased from 78 in 1970 to 348 in 2004 [26]. Floods and their impacts are likely to increase in the future due to urbanization, land-use change, lack of regulations, and poor preparedness efforts [27]. Undoubtedly, South Asian countries will bear the major brunt of climate change.

People move to cities presuming to be safer against climate-related natural disasters in Pakistan [28]. However, overpopulation, congestion, and haphazard urban growth make urban areas dangerous as compared to the countryside. Pakistan's Planning Commission has also acknowledged that rapid urbanization and climate change reinforce the negative impacts of each other [24].

(b) Unregulated urbanization: in the last decade alone, low and middle-income countries faced 53% of global disasters yet they suffered 93% of the fatalities [29]. Such significantly polarized impacts on the developing world are large because of unsafe and unregulated urban development [30], which leads to natural disasters. In fact global economic losses from natural disasters estimated at USD 232 billion from 2000 to 2020 [31]. The global urban population is expected to rise from the present 50 to 66% by 2050. It is expected that a 90% urban growth would take place in Africa and Asia [32], where South Asia will top the list with major capital investment to build new houses to accommodate the burgeoning population. This housing growth will take place in cities with poor capacity to ensure risk-sensitive construction, putting the lives of vulnerable and poor people at

higher risk to natural hazards [30]. The referred literature re-emphasizes the fact that haphazard urbanization is a major challenge in Pakistan.

(c)　Housing shortage: research suggests that globally 1.4 million people are moving into urban areas every week. To meet the rising housing demand, humankind will build 1 billion new residential units by 2050, which is more than the houses built in the entire history of mankind [33].

The State Bank of Pakistan estimated an urban housing shortage of 4.4 million in 2015. The five largest cities in Pakistan will have 78% of the total housing shortage by 2035. The Framework of Economic Growth (FEG) and Vision2025 explicitly acknowledge the housing crisis in Pakistan, to be mainly due to horizontal urban growth. For example, FEG provides a comparative example of Dubai and Pakistan. Figure 1 depicts that in Dubai, 0.2 million people live in 1 km$^2$, whereas in Pakistan the corresponding figure is merely 6 thousand. This shows that Dubai's urban density is 27 times greater than that of Pakistan's [34]. According to the FEG, the reason for low urban density in Pakistan is the adoption of the "garden city" approach in the early years of independence. The absence of tower cranes, strict land regulation, and zoning policies stifled vertical urban growth and development of downtowns while allowing Pakistani cities to develop large suburban sprawls [22]. Unfortunately, this policy continues unabated.

The absence of high-rises does not mean that they are unfeasible in Pakistani cities. Such mixed-use high-rise development was a norm until the sixties when the "garden city" paradigm promoted single housing development [22]. In fact, multi-story buildings are commonly available in big cities, demonstrating their commercial and structural feasibility.

While acknowledging the housing shortage in Pakistan, the successive governments devised plans like the FEG and Vision2025 to address the shortage. Lately, the Pakistan Tehreek-e-Insaf (PTI) government has also made an ambitious strategy to increase the availability of residential units in the country.

(d)　Diminishing social capital: social capital is defined as trust, connectedness, and teamwork in a community. Unfortunately, inadequate and dilapidated public spaces in Pakistani urban areas such as town squares, community centers, theaters, playgrounds, forums, shopping centers, and libraries are the reason for reduced social capital in the country. FEG understood this important need in urban development as it desired the availability of more public spaces while duly considering the context of high-rise and mixed-use construction [34]. Regrettably, the entire plan could not be materialized with the change of government in the year 2013.

(e)　Inadequate Spatial Planning: Disproportionate and outdated zoning laws have exacerbated the rational use of urban land for residential, commercial, and industrial needs. For instance, the best planned city of Pakistan, Islamabad, has 55% of the land reserved for residential use, whereas only 5% for commercial activity, which leads to unplanned and haphazard urbanization [34]. Similarly, the big cities in Pakistan, like Karachi, Lahore, and Faisalabad face exponential growth in slums and *katchi abadis* (shanty towns) without any basic municipal facilities. Such unplanned growth will ultimately lead to unsustainable and retarded economic growth [24].

(f)　Ineffective building by-laws: enforcing building codes and land use planning to deal with mass disasters is a prerequisite [35]. Unfortunately, they are not implemented in many developing countries including Pakistan. For instance, poorly built buildings were severely damaged during the 2005 earthquake in Pakistan in which thousands of people lost their lives under the collapsed buildings [36]. Similarly, a 7.8 magnitude earthquake in Nepal in April 2015 took 9000 lives and demolished built infrastructure [33]. In developed countries like Japan, earthquakes of a similar magnitude normally cause lesser damages due to resilient building infrastructure. The country's well developed national laws based on scientific research, engineering analysis, a framework for certification, inspection, professional and workforce training, building finance, and insurance have reduced the risk of natural hazards.



(g)  Urban water scarcity: industrialization, urbanization, and population growth, coupled with inefficiencies in water use, leads to groundwater depletion and the declining quality of surface water. Climate change aggravates these pressures [37]. In fact, water scarcity is a global issue where 78% of the world population will be facing physical and economic water scarcity by 2025 [38].

Pakistan's per capita water availability has already reduced from 5300 m$^3$ in 1947 to less than 1000 m$^3$ in 2016 [39]. Approximately, 120 million people in Pakistan face severe water scarcity during at least part of the year [40].

National estimates suggest that in Punjab, the groundwater table has gone down by 15 to 20 feet in the last five to six years, whereas in Khyber Pakhtunkhwa, it has been going down by 6 to 21 inches every year [41]. Droughts, unexpected water supply interruptions, or dilapidated networks may further jeopardize the water resilience of urban areas, which may trigger social and ethnic tensions especially in socially and ethnically diverse cities like Karachi and Peshawar. Considering the increasing population and water disputes with India and Afghanistan, per capita water availability will shrink further. These are the neighboring countries from where Pakistan gets the most of its freshwater inflows.

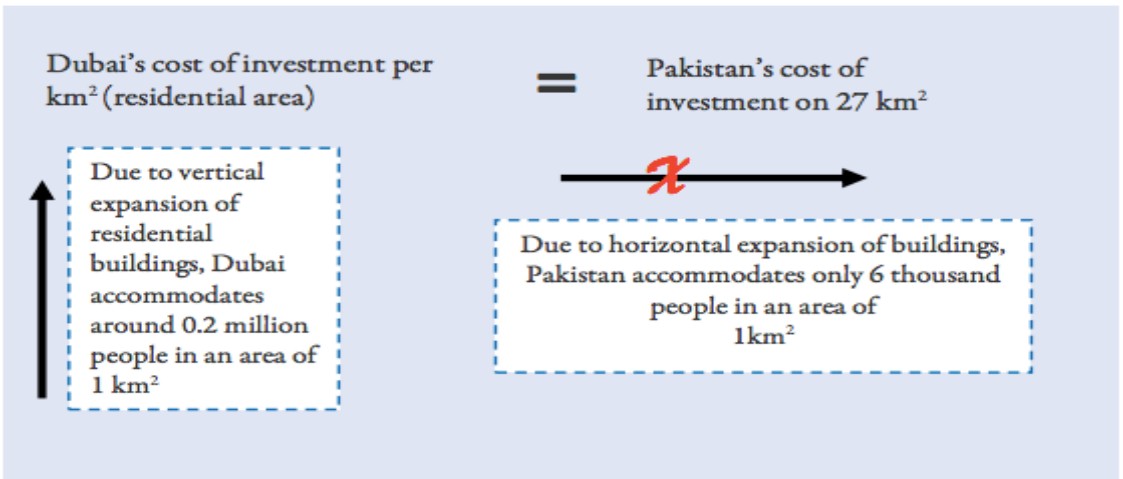

**Figure 1.** Population density comparison of Dubai and Pakistan (Planning Commission [34]).

In short, these are only some of the major challenges Pakistani cities face at present. They will keep mounting by embracing the surrounding localities and permanent migrants from rural areas into their ever-increasing geographical boundaries.

*3.3. Urban Resilience Discourse*

For the sake of convenience, discourse on urban policy and planning is divided into two broad categories. The first is related to the nationally or indigenously drafted plans and policies, and actions for achieving resilient urbanization, whereas the second will include global urban plans, policies, and actions that are binding upon the member nations or they have adopted them voluntarily for attaining urban resilience.

3.3.1. National Urban Discourse

Local governments carry out urban governance and management in Pakistan. They are protected under Articles 32 and 140-A of the constitution. All four provinces of Pakistan have their respective local-government-enabling legislation and ministries responsible for the implementation of urban policies. There are 2055 urban councils in Pakistan, which include one city district, four metropolitan corporations, 24 municipal corporations, 280 municipal committees, 148 town councils, and 1598 urban

union committees. Additionally, 43 federally administered Cantonment Boards also function as local government in urban areas of Pakistan by providing allied municipal services to their residents [42].

Except for an urban unit in the Punjab that was established in 2006, and in 2012 converted into a private company for Urban Sector Planning and Management Services Unit Ltd., there is no National Urban Policy Institute in Pakistan [22,42]. Rapid urbanization, bereft of an overarching urban policy, has stifled economic growth in Pakistan. This is probably the reason that the country and municipal-level governments are left to fend for themselves to reduce the damages associated with catastrophes [7]

Nevertheless, the Pakistani government has made urban development plans and policies either exclusively or in combination with the overall national development plan. These are discussed especially in relation to urban development planning.

(a)　Framework for Economic Growth: in the year 2011, the Planning Commission of Pakistan prepared the Framework for Economic Growth. In this document, reasonable emphasis was made on making cities creative. The FEG aimed to spur economic growth while considering the cities as engines for economic productivity. The idea for creative cities was for promoting mixed-use activities, encouraging energy efficiency, facilitating vertical growth, privatizing unproductive state-owned land, encouraging foreign land developers to compete in the Pakistani real estate market, and focusing on research and development in low-cost energy-efficient construction techniques [34].

FEG also elucidated some elaborate provisions for promoting the housing sector in Pakistan. For instance, it advised modernizing the land registration system in a centralized database, establishing a housing database such as the price index, the access index in assistance with national organizations like the National Database and Registration Authority (NADRA) and the Federal Board of Revenue (FBR), registration of property dealers, releasing unproductive state land, curtailing the growth of slums and encouraging high-density, mixed-use urban development [34]. Even though never implemented, FEG remained a benchmark for policymakers in the following years.

(b)　Vision2025: the next major planning document of the Planning Commission was Vision2025. It was introduced in the year 2014. It acknowledged the serious urban challenges in Pakistan. The Vision2025 enlisted many measures for transforming Pakistani urban areas into the most advanced and creative cities so that they can be on par with the cities of the developed world. For instance, Vision2025 proposed creative, eco-friendly, and sustainable cities. The government also envisioned the availability of efficient mass transit systems, better security, zoning laws for 'mixed-use' areas, vertical rather than horizontal growth, meeting housing shortage, the provision of adequate municipal services, developing pedestrian-friendly streets, the digitization of the land registration system, maintenance and protection of heritage sites and digitally intra and interconnected cities for real-time data sharing so that cities in Pakistan be smart and creative in the future [24].

All measures were enlisted without any detailed implementation planning except mentioning the presence of an Urban Planning Unit at the Ministry of Planning, Development & Reforms. Since urban councils are provincial subjects, except for the federally administered 43 Cantonment Boards and the capital Islamabad, it is yet to be ascertained if any real urban development work or consultation has been undertaken with the provincial governments with regard to the implementation of Vision25.

(c)　Climate Change Policy: the Ministry of Climate Change introduced this policy in the year 2012. The ministry confessed the serious impacts of climate change on urban areas. It proposed several policy measures for climate change adaptation and mitigation. In an urban area, town planning was made a prerequisite for the adaptation to climate change. The policy also desired low-carbon emissions by human settlements with properly managed fuel and energy consumption [43].

For mitigating climate change impacts on urban areas, respective municipal governments

will introduce changes in town planning and building systems. For achieving this purpose, the municipal bodies will build wastewater treatment plants, modernize solid waste management, cut carbon footprints via updated town planning, design zero-emission buildings through renewable energy technology, ensure "land use planning", encourage vertical rather than horizontal urban expansion, undertake the mapping and zoning of land for industrial areas, and make the installation of solar water heaters mandatory for commercial and public buildings [43].

However, the implementation of the policy is a big challenge, which has already been acknowledged by the ADB in its report on the climate change profile of Pakistan in the year 2017 [44]. Further research is also needed if urban policy measures from the climate change policy section were ever given any serious thought. Whether municipal bodies of various cities from different provinces were ever taken on board or any consultation was held for implementing the policy recommendations.

(d) Urban Climate Change Resilience Trust Fund (UCCRTF): the Asian Development Bank in collaboration with Oxfam and the Omar Asghar Khan Foundation launched a project of urban financing partnership facility for building resilience at the community level in different cities of Asia. Abbottabad and Sialkot from Pakistan are set to benefit from the collaborative project, which aims to find the most vulnerable wards of the two cities for developing their resilience by selecting the members from the local community. The Community Stakeholders Group (CSG) will be composed of women, youth, and persons with disabilities for leading and implementing the project. A workshop has already been held in Abbottabad municipality on 6 February 2019 where organizers introduced goals, outputs, and work plans of the project execution.

(e) World Bank-funded Sindh Resilience Project: this scheme aims to deal with floods and drought for building water resilience in the province. The project is for building flood embankments along major riverine canals, constructing small recharged dams for protecting communities from torrential and flash flooding, and developing the capacity of the Sindh Irrigation Department for equipment upgrading, and river morphological studies. Unfortunately, in this project, the researchers could not find a single urban specific investment in any city of the province for building urban resilience in the most urbanized province of Pakistan. The reason for mentioning this project in this urban-specific literature is that this is the only project in Pakistan which mentions the term "resilience building".

(f) Urban Sector Planning and Management Services Unit Pvt. Ltd. (The Urban Unit): the Punjab government established this knowledge-based private organization in 2006. Subsequently, the provincial government renamed this Project Management Unit and registered it with the Securities and Exchange Commission of Pakistan in the year 2012. The Unit is adequately equipped with financial and human resources. A team of 400 having expertise in all sectors of urban planning and management are working in the organization for the urban policy area.

Its mandate is to give policy advice and provide services to the public and private sector organizations in the field of housing, urban planning, urban transport, solid waste management, water and sanitation, urban economics, municipal finance, institutional development, capacity building, and urban services delivery improvement.

The unit has extensively published a database on urbanization and urban issues especially in the largest province of Pakistan i.e., Punjab. Urban discourse on the Urban Unit website covers a wide range of topics like green spaces, transportation, and Pakistan's urban growth data in recent decades. It also publishes an Urban Geographic journal and organizes consultative discussions with national stakeholders on the latest issues like smog in Lahore.

(g) Naya Pakistan Housing Project: the Pakistan Tehreek-e-Insaf (PTI) government has initiated an ambitious plan to provide housing facilities to the urban residents at a reduced cost. The initiative, if executed as planned, will certainly reduce the housing deficiency in the urban areas. The project will not only increase the social resilience of urban residents but also reduce the slums or

*katchi abadis.* However, the housing plan table of the proposed scheme shows that it has chosen the existing model of horizontal urban expansion rather than opting for the vertical path.

The table (Figure 2) reveals that one-unit houses will be maximum followed by additions to existing stories, ground+3, and midrise. Considering the quality of construction prevalent in Pakistan where multi-story buildings keep collapsing every other day, adding further stories to existing buildings will not only jeopardize the lives of existing residents but also endanger the future inhabitants. High-rise has been given the last priority in the plan.

(h) Orangi Pilot Project (OPP): it is a non-governmental organization (NGO) established in Karachi in the year 1980 by Dr. Akhtar Hameed Khan [46]. Dr. Hameed was an eminent philanthropist, who selflessly worked for uplifting Orangi Town, one of the largest slums of Pakistan in Karachi. This slum is a cluster of 113 low-income settlements, housing 1.5 million people. The self-supported organization aimed to uplift the lives of the slum residents through five programs such as low-cost sanitation, housing, health, education, and credit for micro-enterprise.

The organization has three branches. The first is the Research and Training Institute, which manages low-cost sanitation, housing support, education, water supply, and women's saving programs. From this platform, the project has succeeded in approving land tenure security to 1063 goths (villages) by mid-2010 and provided a loan to 100 houses on an annual basis [47]. The second is Orangi Charitable Trust, which manages a micro-enterprise credit program. The third is Karachi Health and Social Development Association (KHASDA), which runs a health program. The project was immensely successful as it improved the lives of a million residents.

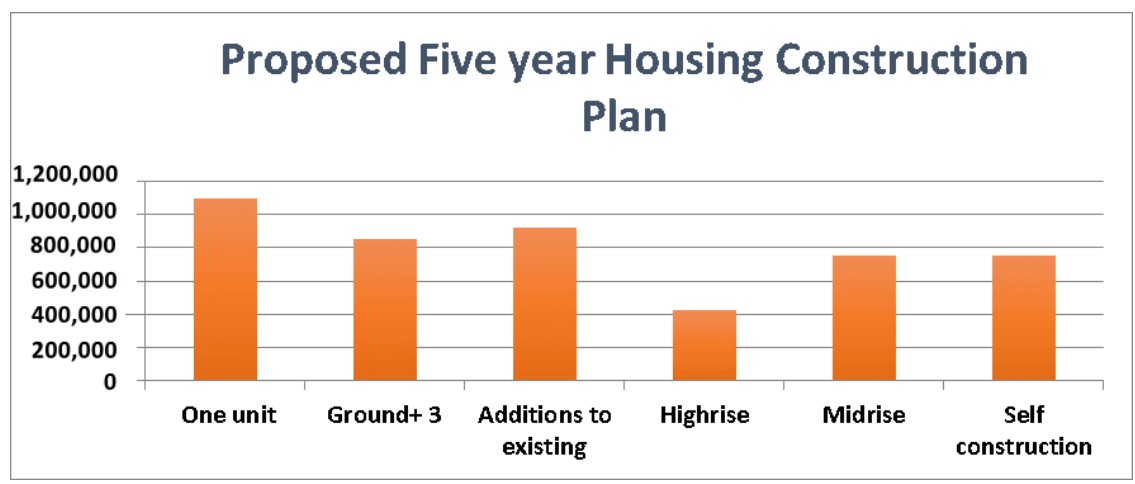

**Figure 2.** Pakistan Tehreek-e-Insaf (PTI)'s housing plan (Naya Pakistan Housing Scheme 2020 [45]).

3.3.2. Institutional Framework against DRR

Municipal and local bodes in urban areas are directly responsible for the management of urban areas. However, there are some other federal and provincial government institutions which supplement the functioning of these urban governance bodies in disaster risk reduction.

Pakistan Meteorological Department (PMD), which was established in 1947, provides information on the early warning of natural hazards including drought, flood, tropical cyclone, tsunami, seismic activities, and advisory services in the fields of planning and development, town planning, and infrastructure. It has four divisions based in different cities of Pakistan. However, two departments can contribute to urban resilience. These departments are flood forecasting division at Lahore and the National Seismic Monitoring and Tsunami Warning center at Karachi [44].

National Disaster Management Authority (NDMA) was set up under the National Disaster Management Act 2010 for laying the guidelines and formulating plans, strategies and programs on disaster risk reduction (DRR). The entire NDMA structure is composed of national,

provincial, and district level bodies for DRR at their respective levels. The Ministry of Climate Change is the apex body that creates linkages for Climate Change Adaptation (CCA) and DRR [48].

Civil Defense, established under an Act in 1952, is another organization that works under the Ministry of Interior for the provision of defense and emergency services especially in times of enemy attack. It is a semi-trained workforce which is made active in times of crisis [49].

Global Change Impact Studies Centre (GCISC) is an autonomous institution dedicated to climate change impact studies. However, its work on urban resilience is little revealed and understood [50].

These government bodies are working independently for the overall safety of the national population regardless of their location and specific objectives. They are not directly involved in building urban resilience.

### 3.3.3. Globally Adopted Urban Resilient Policies and Plans

As a responsible member of the international community, Pakistan has participated, contributed, and adopted almost all global plans and treaties on urban sustainability regardless of these documents that were drafted at the global level like the United Nations platform, continental levels like those of the ADB or regional level like those at the South Asian Association of Regional Conference (SAARC). The Pakistani government has wholeheartedly accepted them for achieving all enshrined goals for uplifting the living standards of its citizens. The country adopted and implemented the following global urban specified plans.

(a)     Sustainable Development Goals (SDGs): SDG 11 is for "making cities and human settlements inclusive, safer, resilient and sustainable". This goal extensively covers all aspects of resilient urban development. For example, it desires affordable housing by reducing slums, sustainable and affordable urban transport, reducing physical and economic losses due to disasters, paying attention to air quality, solid waste management, and adopting and implementing the Sendai Framework for Disaster Risk Reduction (SFDRR) 2015–2030. However, empirical research is needed for the actual implementation of SDG Number 11 and the Sendai Framework for Disaster Risk Reduction 2015–2030 in Pakistan.

(b)     Sendai Framework for Disaster Risk Reduction 2015–2030 (Sendai Framework) it is a sequel to the Hyogo Framework for Action (HFA) 2005–2015, which was an agreement for a resilient and sustainable development agenda. The UN General Assembly duly endorsed the Framework in 2015 at its Third World Conference on disaster risk reduction. It proposed seven global targets, four priorities for action, and implementation guidelines for the Sendai Framework [51].

Pakistan has adopted the SFDRR to achieve global targets such as reducing global disaster mortality by 2030, reducing the number of affected people, reducing economic losses due to disasters, enhancing international cooperation among developing countries for disaster risk reduction, and installing multi-hazard early warning systems and disaster risk information.

Pakistan Disaster Management Authority (PDMA) is responsible for the coordination and implementation of the framework in consultation with all stakeholders. A national consultative conference for localizing SFDRR in Pakistan was held in Islamabad in collaboration with the United Nations Development Program (UNDP) and UK Aid in February 2017 [52].

### 3.4. Global Risk Reduction and Urban Resilience Models

We read some popular resilient city models and frameworks from different research papers and reports of international organizations for their global effectiveness and practical applicability in Pakistan. These frameworks and models can be good templates for other countries and local governments facing urban challenges. Some of them are described below:

(a)     Resilient qualities: resilient cities usually have some powerful characteristics such as being reflective, robust, redundant, flexible, resourceful, integrated, and inclusive in their systems.

Each one of them is explained below.

Reflective: institutions and their allied stakeholders keep learning from experiences with an adaptive planning mindset so that they can minimize the impacts of catastrophes. They should have dynamic standards for adopting emerging challenges rather than relying on redundant solutions for shocks and stresses.

Robust: city systems are designed and managed in a way to prevent catastrophes and anticipate system failures to enhance the predictability of challenges and the security of cities.

Redundant: this means an extra capacity to meet the untoward demands of city residents if one system becomes redundant. For example, a city can have multiple sources of water or electricity supply. If one system fails to deliver due to unknown reasons, the next should be on standby to prevent interruption.

Flexible: a changing and evolving city will continue adopting alternative strategies both in the short as well as long-term to respond to the changing conditions.

Resourceful: city stakeholders and managers should predict future urban challenges so that they can prioritize, mobilize, and coordinate all kinds of resources in case of extreme events or needs.

Inclusive: an approach in which all urban communities, especially the vulnerable segment of the urban population, are consulted with and are engaged in for building city resilience so that they have a feeling of ownership.

Integrated: investment, decision-making, and city systems should be supportive of each other for a common goal. They should be built in such a way to be in sync with one another and have information and feedback response mechanisms in time of urgency.

(b) The Rockefeller Foundation City Resilience Framework: Arup, in its report on City Resilience Framework, has enlisted eight key city functions that sustain a city's resilience. These include delivering basic human needs, safeguarding the life of human beings, protecting, maintaining, and enhancing physical assets, facilitating identity and relationship among humans, promoting knowledge and information, defending the rule of law, justice, and equity, supporting livelihood, and stimulating economic prosperity [4]. On the contrary, if a city has an unsafe and degraded environment, conflicts, deprivations, insecurity, or ill-health, it is considered as not resilient and extremely vulnerable to shocks.

The Arup report on City Resilience is a comprehensive document based on collected data from cities across the continent having diverse capabilities and resources to cope up with the disasters and which have faced a catastrophe in recent years. The foundation has developed a City Resilience Index, which has four broad categories of City Resilience Index, divided into 12 goals and subdivided into 52 indicators followed by 156 variables.

(c) "Crunch Model" or commonly called "Pressure and Release Model": this model was developed by Oxfam. It helps in understanding and reducing the disaster risk. The model at Figure 3 indicates that vulnerability (pressure), which is endemic in socio-economic and political processes, has to be dealt with (released) so that disaster risk can be reduced and the resilience of the urban areas is amplified.

According to the disaster crunch model, a hazard is an unexpected event, which affects vulnerable people. When two elements i.e., hazard and vulnerability, join in tandem, they influence marginalized people by bringing disaster. A hazard cannot be a disaster if it struck a resilient population. Likewise, a highly vulnerable community can stay safe from a disaster if a triggering event like a danger stays away from the population [53]. Hence a vulnerability pressure, which has roots in socio-economic and political processes has to be addressed and released so that the risk of disaster can be minimized.

The original model is not much different from the latest one, which is reasonably brief. According to the original model, people are vulnerable if they cannot forecast, withstand, and recover from a disaster.

The two-dimensional model has a vulnerability progression and hazards as major components.

The root causes of vulnerability are limited access to power, structures, and resources with weak political and economic systems. Dynamic pressures like the lack of effective local institutions, training, and investment, as well as population growth, rapid and haphazard urbanization, deforestation, and soil degradation, merge with dangerous locations, dilapidated buildings and infrastructure, lack of disaster preparedness, and endemically prevalent diseases to bring disasters. These factors are combined with opposing hazards like earthquakes, fast winds, cyclones, hurricanes, landslides, drought, and volcanic eruptions to damage lives and livelihoods [53].

(d) Three levels of city resilience: the Asian Development Bank proposed this model in its report on climate change resilient cities. According to the ADB, a city's functional systems can bear shocks and stresses whereas nonfunctional systems cannot. In functional cities, frequent stresses do not impact on peoples' and organizations' everyday decision-making. Moreover, peoples' and organizations' capacity to fulfill their aims is continually supported by the cities' institutional structures [3].

(e) Risk reduction (resilience) model: it was proposed by Mehrota [54] and amended by UK Aid by adding a resilience dimension. According to the model when any stress or a challenging situation arises in an urban area resulting from natural disasters, drought, smog, food shortage, a concomitant increase in refugees, crimes, and criminals, they will test the residents' vulnerability and resilience. Urban machinery such as institutions, civil society, public, and local action groups can reduce both acute and chronic urban challenges.

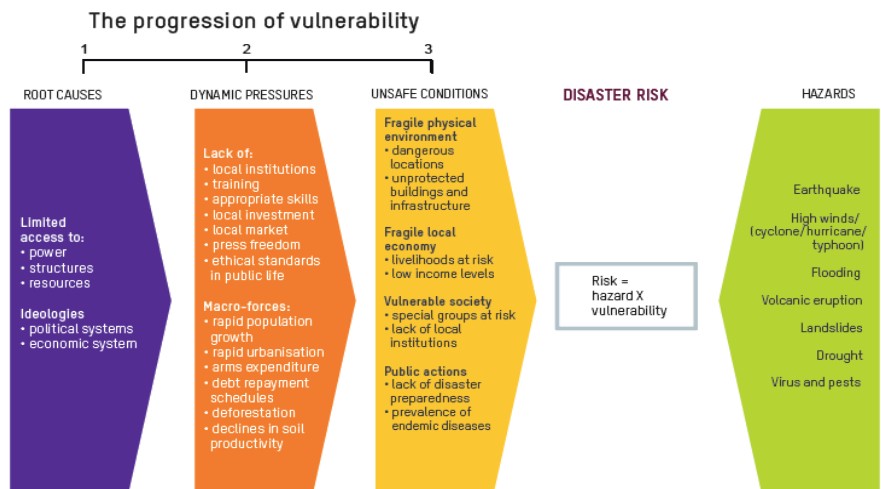

**Figure 3.** Original crunch model (Oxfam [53]).

*3.5. Disaster Risk Reduction and Relief Projects—Global Examples*

Many global organizations like the United Nations, US Aid, and the Department for International Development (DFID) are engaged in relief and rescue operations all around the world. Their purpose is to help developing countries and communities to recover from disasters. These organizations have accumulated valuable experiences and research data while working in different geographical and ethnic areas. Their experiences and project models are highly useful as they can be implemented anywhere in the world with minor modifications as per local needs. Some useful examples of their relief projects are given below:

(a) The Katye Neighborhood Upgrading and Recovery Program: a devastating earthquake of 7.0 magnitude struck Port-au-Prince, capital of Haiti, on 12 January 2010. The earthquake affected 3 million people [55]. Ravine Pintade was among the hundreds of informal settlements badly affected by the disaster. Almost 90% of the residents were affected, whereas infrastructure was badly damaged and became inaccessible [56]. USAID and the Office of the US Foreign

Disaster Assistance (OFDA) funded project was initiated for providing relief and recovery for the residents of Ravine Pintade. The goal was to meet the basic humanitarian needs of those affected and displaced by the earthquake to provide safe and habitable neighborhoods along with providing essential services [57]. The Katye pilot program aimed to ensure expert engagement, community participation, and coordination among government agencies to face natural disasters. The project was different from others in a way that it directly engaged with the affected households to rehabilitate their original neighborhoods rather than shifting them to camps and Greenfield construction. Within ten months of starting in November 2010, it succeeded in providing health, shelter, livelihoods, debris removal, water, sanitation, and hygiene (WASH) services. It was evaluated that the project generated substantial trust and mutual understanding between the communities and implementing agencies. Thus, Katye was a notable success [57].

(b)  Barrio Mio: Barrio Mio (My Neighborhood) was also a disaster risk reduction (DRR) project. It was funded and managed by Project Concern International (PCI) and OFDA, respectively. The project was carried out in 17 vulnerable settlements of Mixco city of Guatemala in three phases starting from 2012 [58]. The first phase was from October 2012 to March 2015, the second one from April 2015 to October 2017, and the third and last phase was expected to end in September 2020. The objectives of the project were to identify, pilot, and scale solutions to strengthen high-risk urban informal settlements, improve emergency response to disasters and convert vulnerable areas into safer, healthier, and resilient neighborhoods. It joined 40 stakeholders ranging from national and local governments, academic institutions, the private sector, and community participants. The strategy was to change, influence, advocate, and reduce the risk for people from the lower scale to a higher scale of governance.

The basic approaches of Barrio Mio for addressing the vulnerability of precarious urban settlements and their inhabitants were including risk and vulnerability, participatory mapping/enumerating, identifying and piloting innovative shelter, water and sanitation retro-fitting solutions, women empowered groups, disaster response, supporting and strengthening the institutional capacity of local councils [58]. Initiated in 17 precarious settlements, the project will expand its operation by including more settlements for strengthening their resilience to disasters.

The foregoing sections comprehensively discussed the national urban discourse and international urban resilience models and practical examples. The following section will carry out a comparative analysis of the national and international urban discourse.

## 4. Comparative Analysis

Almost all mentioned national development plans desire urban development. The kind of development includes upgrading city infrastructure, provision of civic and municipal services, developing and continuously upgrading urban zoning plans, and improving urban air quality. However, none of them set resilient urbanization as a goal for mitigating urban challenges. The term urban resilience is scarcely used in all these plans and policies. This is the biggest shortcoming in urban literature considering the serious and widespread urban challenges in Pakistan.

Nevertheless, the absence of terms in national development literature does not mean that the government has ignored building urban resilience. The federal and provincial governments have taken many urban development initiatives in recent years in various urban areas to modernize cities and improve the livelihood of urban residents. For instance, the federal government in collaboration with the Punjab government has developed a metro bus for the twin cities of Islamabad and Rawalpindi. Similarly, urban mass transport systems have also been made operational in Lahore and Peshawar whereas, in Karachi, the green line urban transport system is in the final stages of construction. Another major milestone in the mass transit system, known as the Orange Railway line, has also been made operational in Lahore in collaboration with the Chinese government. Regardless of these efforts, the public mobility demand far outweighs the supply. It shows that much more needs to be done on every front to combat urban poverty and improve the miserable living condition of urban residents.

Moreover, the national urban discourse confirms that the government and municipal agencies certainly desired urban resilience even if they did not explicitly mention the term resilience in their development plans. However, cities could have benefited more if disaster resilience and risk reduction were included in development plans.

For example, the federal and provincial governments could have directed their respective municipal bodies for identifying vulnerable communities or addressing their susceptibility. Most immigrants in cities are living in low-lying dangerous localities around the cities, and these slums lack basic urban amenities like clean water, sewerage system, paved streets, and health care facilities. By engaging with the slum communities in line with OPP, Katye neighborhood project, and Barrio Mio, the municipal bodies could have lessened the burden on strained civic services.

Unfortunately, national development plans rarely desired or proposed community participation. The community is a strong stakeholder. Empowering and including the public in urban development plans will not only help the government to swiftly and efficiently implement its urban reform programs but also promote a sense of ownership of the towns. For instance, the Phnom Penh Water Supply Authority (PPWSA) shows that communities can be effective planners and regulators which is an essential aspect of successful and resilient urbanization [59]. The success of PPWSA in the provision of clean and safe water to more than a million people in Phnom Penh is a result of the combination of public sector activism and community-level participation [59].

Likewise, regular monitoring is necessary for the execution of planned projects. However, one can easily guess from the ubiquitous urban challenges in Pakistan that these rosy plans might have been ignored altogether or implemented piecemeal and haphazardly, leaving the cities to deteriorate further. In Pakistani cities, poorly executed projects like the Peshawar metro show that monitoring and evaluation mechanisms are missing.

The Table 1 has a summary of the entire plans, policies, frameworks, and models already discussed in the previous sections. The section ends with a discussion of the table.

It is apparent from the above table that Pakistani urban discourse is mostly development oriented. Although it does promote creative and eco-friendly cities to build energy efficiency, saving precious land by going vertical, empowering the community, developing mass transit systems and addressing housing shortages, however, it drastically differs from global urban resilience building models as cited in the table. For instance, global resilience discourse uses terms like disaster anticipation and risk reduction. It also proposes developing alternatives to urban civic services such as water supply networks, electricity, and gas. Likewise, releasing socio-economic pressures, and taking risk reduction measures like identifying and removing dangerous buildings, eradicating endemic diseases, and controlling population growth are common terms often used in global urban resilience building literature. Such types of efforts are lacking in national urban development initiatives despite Pakistani cities facing frequent climate-related and other disasters.

Lastly, national urban discourse is completely missing in disaster risk reduction jargon. On the contrary, international discourse is bulky on terms like disaster risk reduction, disaster anticipation, risk identification, and mapping vulnerable communities. The Pakistani government must consider this lacuna in future urban planning discourse.

In short, the national urban literature uses a top-down and passive approach whereas the global resilience-building discourse conforms to a bottom-up and active approach. For instance, the former uses the term community empowerment while the latter desires community engagement. A passive approach cannot be sustainable in the long run as it has to rely on federal or provincial governments for financial and technical support. In an active approach, the cities and their allied institutions have to develop their own intrinsic and inherent abilities to run urban affairs. Because strong and dynamic institutions are vital for resilient and sustainable urbanization, it is important to minimize the impacts of the extensive human footprint on the planet and environmental degradation so that our cities can be amenable to productive living [60].

**Table 1.** Urban challenges and planning discourse.

| Urban Challenges in Pakistan | Climate Change, Unregulated Urban Development, Housing Shortage, Diminishing Social Capital, Inadequate and Faulty Spatial Planning, Insufficient Building Bye-Laws, Water Scarcity, Poor Air Quality | | |
|---|---|---|---|
| **National Urban Plans and Policies** | **Summary of Proposed Actions** | **Global Urban Resilience Models** | **Summary of Proposed Actions** |
| (a) The Framework of Economic Growth <br> (b) Vision2025 <br> (c) Climate Change Policy <br> (d) Naya Pakistan Housing Program <br> (e) Urban Climate Change Resilience Trust Fund <br> (f) Sendai Framework for Disaster Risk Reduction 2015–2030 <br> (g) Sustainable Development Goals <br> (h) Urban Unit, Lahore <br> (i) Orangi Pilot Project, Karachi | Creative and eco-friendly cities, mixed-use activities, energy efficiency, vertical expansion, mass transit systems, housing shortage, rescue services, data sharing, waste-water treatment plants, land mapping and zoning, community empowerment, slums reduction, air quality, a micro-credit program | (a) Qualities of a Resilient City <br> (b) Rockefeller Foundation (City Resilience Framework) <br> (c) "Pressure and Release Model" <br> (d) ADB's Three levels of city resilience <br> (e) Post-Disaster Recovery Efforts <br> (f) The Katye Neighborhood Upgrading and Recovery Program <br> (g) Barrio Mio | Reflective cities, adaptive planning, innovative solutions against shocks and stresses, disaster anticipation and prevention, urban security, additional water and electric supply capacities, mobilization and coordination of existing resources, community engagement, promoting a sense of ownership, delivering human needs, knowledge promotion, releasing socio-economic and political pressures, reforestation, identification of dangerous buildings, eradicating endemic diseases, increasing income levels, controlling population growth, dynamic and strong urban institutions, risk identification, participative mapping, piloting innovative shelter, water and sanitation retro-fitting, women empowered groups. |

## 5. Recommendations

Pakistani cities are beset with immense urban challenges. Therefore, developing urban resilience is paramount to avert disasters. From these models, the researchers have found the "Pressure and Release Model" relevant for the country. The model mentions the majority of urban issues Pakistani cities face at the moment. For example, limited access to power, weak political and economic systems, high population growth, ineffective local institutions, haphazard urbanization, deforestation, and endemically prevalent diseases are common urban issues in most Pakistani cities. These pressures and their ilk are either specifically mentioned in the "Pressure or Release Model" or they have been indirectly referred to for their resolution. Indeed, all these issues seriously afflict the marginalized segment of urban residents. Once urban pressures are released, the fringe urban residents will be able to bear shocks and recover quickly from extreme events.

However, specifying the "Pressure and Release Model" does not mean that other resilience models do not relate to Pakistan or the researchers have found them irrelevant in the context. For instance, achieving resilience qualities like cities being reflective, redundant, flexible, resourceful, robust, integrated, and inclusive are equally important. The urban planners and municipal managers must consider these aspects in the development and management of cities. Similarly, the risk reduction model of UK Aid will be considered implemented in urban areas if the "Pressure and Release Model" is adopted in its entirety. With the implementation of the risk reduction model, cities will also achieve most of the resilient qualities as already mentioned in the second sentence of this paragraph. In fact, most of these models are interlinked in the sense that they aim to build sustainability, reduce risk, and minimize the impacts of extreme events. Achieving the objectives of one model means that the goals of other models have also been realized.

Furthermore, several risk reduction examples and precedents have been found to be extremely useful in addressing community vulnerability in urban areas around the world. These precedents are mentioned in the paper and recommended for implementation in Pakistan at suitable places depending upon the available financial resources and physical modalities. For example, the negative consequences of haphazard urbanization, natural hazards, as well as physical and psychological insecurities can be prevented by strengthening urban residents' capacity to anticipate, resist, cope with and recover from natural hazards [61]. Risk management in urban areas may be considered as a high priority for the government, considering the role of cities in the national economy, and centers of intellectual learning, businesses, and financial activities [62]. It can be achieved by building risk-reducing infrastructure and services, such as drainage systems, waste collection, sanitation, emergency services, and health care facilities [35]. Enhancing individual capabilities, expanding services networks, connecting communities with national institutions along with taking soft measures such as new regulations, technology, and information systems, and social networks are equally important. Engaging multiple stakeholders from various socio-economic groups and economic interests across different sectors i.e., government, businesses, civil society, and academia for transformative changes may also be considered as real action plans. Understanding the connected systems located beyond the city boundaries to interact economically, physically, ecologically, and politically for an effective and coordinated response for building urban resilience.

Furthermore, inclusive future planning with today's needs such as water supply and urban drainage will bring future scenarios into current decision-making. In addition, tapping the local expertise is more important. For instance, the involvement of academics and key-informants in bringing quality engagement and long-term adaptive planning capacity can also be considered. Likewise, focusing on vulnerable communities by identifying, empowering, educating, and engaging them in decision making will help them and cities to be more resilient [3].

The World Bank Group and the Global Facility for Disaster Risk Reduction (GFDRR) are partnering with governments, the private sector, and civil society for building resilient buildings. The program studies the best global building construction practices such as those of Japan while emphasizing the cultural, economic, and social factors for developing region-specific building codes for a resilient future.

For example, in Northern Pakistan, the *dhajji dewari* approach (a timber and stone earth construction practice developed over centuries) can be a cost-effective and resilience-building practice for local needs. Many similar construction techniques can be studied and implemented at the local level for resilient homes [33]. Such affordable and resilient homes will minimize the impacts of disasters on urban areas.

In addition, when it comes to owning a house, affordability is the major issue for low-income families. For helping such households, private developers should allocate at least 50% of the land for poor communities. The remaining land can be equally divided between medium-income and high-income groups at actual and inflated prices to compensate for the loss of providing land to poor families at discounted rates [46].

Many well documented resilience-promoting tools have been planned over the years. For example, facilitating preparedness, effective planning against challenges, and raising public awareness are some of its examples. The role of the international, national and local governments, civil society, and the private sector, etc., is crucial for achieving urban resilience. The public availability and effective communication of data increases the planning and preparedness prospects. Likewise, financial readiness is equally vital along with political commitment. For this, micro-insurance and micro-finance such as catastrophe bonds for providing liquidity in times of crisis; and country-level finances will help reduce public sector stress [35].

Last but not the least, cities should adopt an iterative, inclusive, and integrated planning process for urban analysis. Understanding how the city works and analyzing its present population pressure and future population and economic growth projections, and vulnerability analysis are key for building robust and resilient cities. Identifying the most vulnerable segment of the urban population especially those residing in low-lying areas, flood plains, and highly congested slums, who will be exposed to urban pressures and have limited coping capacity with which to weather the impending impacts, is critical for their survival in times of crisis [3].

Cities are diverse and dynamic. These recommendations have geographical and chronological constraints. Some of them can be relevant today in a certain city. Tomorrow they might be outdated. Therefore, urban planners and managers keep abreast of the ever-emerging challenges and their allied solutions.

## 6. Conclusions

In spite of witnessing rapid urbanization and the economic potential of cities, Pakistan's economic and development policies are still fixated on agriculture. Soon, the majority of the population will begin to live in cities. Without efficient, effective, and functional systems, cities in Pakistan will be a liability, jeopardizing the economic sustainability of the country. It is therefore imperative for national leaders to prioritize resilient urban development for sustainable economic growth.

The discussed urban literature reveals diverging trends between global and Pakistani urban policies. For instance, global urban resilience models are focused on reducing disaster risk, strengthening institutions, building alternatives to civic services, and promoting disaster preparedness. On the contrary, Pakistani urban discourse is focused on the urban development paradigm. Development without sustainability is short-lived. The country's urban landscape is yet to develop a sound disaster risk reduction plan, which is extremely important considering the urban challenges and rapidly unfolding climate change impacts on South Asia.

Empowering and engaging with the community have different connotations. The former is an end-product whereas the latter is a self-sustaining process for building urban resilience. In Pakistan, Dr. Akhtar Hameed is a pioneer of this innovative method of community engagement in self-development. Unfortunately, the lack of state patronage to adopt his model deprived the urban residents of obtaining better civic services. The same model was successfully implemented in Barrio Mio and Katye neighborhood upgrading and recovery project by the international aid agencies with

astounding success. This proves that such projects could have been implemented in other Pakistani cities via government backing.

Finally, releasing urban pressures is paramount for minimizing the impacts of extreme events. People migrate towards cities to make their dreams come true. However, they live a nightmarish life. Therefore, government and municipal bodies help urban residents live a life full of capabilities. This cannot be achieved without their active involvement.

**Author Contributions:** Conceptualization: L.A. methodology, T.-f.Y. and L.A.; validation, T.-f.Y. and L.A.; formal analysis, L.A.; investigation, L.A.; resources, T.-f.Y.; data curation, L.A.; writing—original draft preparation, L.A.; writing—review and editing, T.-f.Y.; supervision, T.-f.Y.; funding acquisition, T.-f.Y. All authors have read and agreed to the published version of the manuscript.

**Funding:** This research was funded by the China National Key R&D Program.

**Conflicts of Interest:** The authors declare no conflict of interest.

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
