# Peer review of "Resilient Urbanization: A Systematic Review on Urban Discourse in Pakistan"

_urbansci, doi:10.3390/urbansci4040076_

Round 1

Reviewer 1 Report

This paper has been revised and the quality of contents has also been improved. However, more research efforts are needed for further improvement. I have the following concerns.

  1. There are four goals of this review paper (lines 58-61). The second one is “an appraisal of global urban resilience models.” But it introduces various models, without any appraisal.
  2. Authors explain that urbanization increases GDP (lines 105-111). Meanwhile urbanization also causes negative effects on environment and urban life. The urban problems in developing countries is mainly caused by economic problems. For resilient urbanization, authors must provide their opinion on how these conflict can be overcome.
  3. The density comparison between Dubai and Pakistan may bring about a wrong signal, though partly appropriate. The development density depends on many factors, such as revenues from the land, location, stage of economic development, industrial mix, land availability, and so on. The two cases have apparently different background. The current density is reflection of real world. The compact development must be Pakistani version of compact development. In this sense, why the density is lo low in Pakistan cities? Is high-rise residential building feasible in any urban areas?(lines 350-354)
  4. What are the differences between urban policies of Pakistan and authors recommendations to achieve resilient urbanization?
  5. Conclusion is absent which should be based on findings of this paper. Recommendations after short methodology without conclusion is strange format in academic paper.
  6. Authors replied that “total absence of community participation in Pakistan literature”. Is this true? Is there no community participation program or case study in Pakistan?
  7. Should land allocation method for residential land development suggested by authors be accepted simply because the current government puts in practice?
  8. There is no title in figures (Fig. 2, 3, 4 and 6), and Table 1 is a figure, rather than table. In title capital letter and lower letter should consistently be written. Authors should minimize those figures from other report or research.

Reviewer 2 Report

Referee report for the manuscript entitled “Resilient Urbanization: A Discourse on Pakistan’s Urban Policy and Planning”

The manuscript focuses on an interesting topic—resilient urbanization in a developing country context—that is relevant to the journal and academic, policy, and professional audiences. In general, the manuscript that is identified as a review paper reads fine, and while the manuscript has potential to be considered for publication in this leading journal of the field, there are some major structural changes and consolidations must be done.

1) The manuscript is determined as a review paper. This is fine but there is no information on what type of review is this and what methodological approach is used in the review. For instance, is this review a: Argumentative Review, Integrative Review, Historical Review, Methodological Review, Theoretical Review or Systematic Review. Also, if one of these how the reviewed papers are selected and how they are reviewed (which methodological steps are taken)?

2) In general, the paper is well written with a good literature backing. Nevertheless, this base can be improved with the following papers. 

  • Spencer, J. H., & Meng, B. (2019). Resilient urbanization and infrastructure governance: the case of the Phnom Penh Water Supply Authority, 1993–2007. Water Policy, 21(4), 848-864.
  • Srivastava, R. K. (2020). Towards Risk Resilient Urbanization. In Managing Urbanization, Climate Change and Disasters in South Asia (pp. 359-434). Springer, Singapore.
  • Yigitcanlar, T., & Kamruzzaman, M. (2015). Planning, development and management of sustainable cities: A commentary from the guest editors. Sustainability, 7(11), 14677-14688.
  • Córdoba Hernández, R., & Pérez García-Burgos, A. (2020). Inclusive and resilient urbanization in informal settlements. Exemplification in Latin America and the Caribbean. Bitácora Urbano Territorial, 30(2), 61-74.
  • Jansson, Å. (2013). Reaching for a sustainable, resilient urban future using the lens of ecosystem services. Ecological Economics, 86, 285-291.
  • Yigitcanlar, T. (2010). Rethinking sustainable development: Urban management, engineering, and design. IGI Global, Hersey, PA, USA.

3) The manuscript’s methodology section comes at page 16 and does not talk about the review methodology. This is the biggest issue of the manuscript. There is a need for a clear methodology section (in an earlier part of the paper) in detail elaborating what type of review and what steps are undertaken in the study. Also, the way the section is written questions the reader whether this is a review paper. Clarifications are needed on the type of the paper.

4) The comparative analysis section is very brief and limited to provide useful insights. Suddenly paper moves from literature review to policy review here. The authors should provide a clarity on the focus pf the paper, is it a literature review paper or policy review one or both. In all these cases the paper needs to be significantly edited and restructured.

5) In the light of the above the introduction and recommendations/conclusion sections need some editing as well. 

6) Manuscript will greatly benefit from a careful language check. 

7) Keywords should include Pakistan, and you can add up to 10 keywords in total.

8) In general, I've enjoyed reading your paper on Pakistan’s urban policy. Good luck at revision the manuscript. I look forward to reading the improved version. 

Round 2

Reviewer 1 Report

I agree that authors have revised this paper considerably, and the quality of this paper has also a lot been improved. Since I am not familiar wih Pakistani situation of urban resilience, I don't try to criticize the contents of policies or recommendation for resilience urbanization. However, I would like to ask authors to consider the following points. 

It  is not desirable to move whole figures from other researches or reports. Some of them can be explained in text. For this reason, I don't agree that figure 2, 3, 4, and 5 are essential. If essential, source of figures should be written right after the title (in stead of reference number).

Though I pointed out that capital/lower letters should be used consistently in tables, figures, and text(even in the title of this paper!), the problem still exists. For example, words start in capital letters in fig. 6 and 7 while other figures start in lower leter.  In addition there are some spelling errors (ex. Home sapiens?). Check whole text again and correct the errors.   

Author Response

Thank you very much for your review. Please see the attachment. 

Regards

Reviewer 2 Report

The manuscript has been improved significantly.

Author Response

Thank you very much for your review.  As no new revisions were recommended. 

This manuscript is a resubmission of an earlier submission. The following is a list of the peer review reports and author responses from that submission.

Round 1

Reviewer 1 Report

I found it very hard to follow the logic of the paper. What is your purpose of the paper? What is your contribution to knowledge? Could you make them clear in your introduction and conclusion?

There are numerous mistakes in your English.

Section 2: Research Objectives. If it only has two sentences, you should put them into the introduction. 

Please check your citations. Why you have two different citation styles in the manuscript?

Where are your citations for the figures on page 11, 13 and 19?

It is neither a research paper nor a review paper. For a research paper, I do not find any contributions to knowledge nor detailed analyses on the case study. For a review paper, you just listed the concepts and discussions from others.

I hope the authors could take your submission seriously. Your submission took a lot of time from editors and reviewers. Please, at least, check your grammar, citations and figures before submitting it.

Reviewer 2 Report

This paper deals with an important topicresilient urbanization. Though the meaning of resilient city is well explained based on many kinds of international researches, the following points should carefully be considered to improve the quality of this paper. 

First, I don’t find any serious academic contribution of this paper. The contents of this paper do not fit to the title (resilient urbanization).  

Second, this paper simply introduces the meanings of resilient urban models and programs, without deriving meaningful conceptual commons or findings (which can be applied to Pakistan’s resilient urbanization).

Third, figure 3, 4, 5 and 6 are not worth showing. These figures should be explained and compared to derive what model or program is most appropriate for Pakistan case (even though authors insist that they analyzed and incorporated them in conceptual framework).   

Fourth, I don’t agree that showing tables and figures without stating the core of their contents  is an effective way of communication.  Furthermore, all tabels and figure should be assigned with serial number. Is table 1 really a table? I recommend that the figure(Tabel 1) should be in table format with data source. Figure 8 may be deleted. 

Fifth, what is a logical ground of author’s arguementPressure & Release model is most appropriate in page 19 and the Kaytye neighborhood modelin page 20 (there is no figure and table number)? Simple enumeration of models and proposed actions without proper ground cannot support authorsassertion.   

Sixth, community engagement is one of the propesed action among many proposed actions. Why community participation alone is so much important and emphasized? Authors should provide clear evidence/acceptable logic for this discourse. 

Seventh, housing policies for low-income households can be many. There is no logical nexus between resilient urbanization and the allocation method of land by income level (line 705-709). 

Eighth, conclusion is contents of this paper, rather than concluding remarks derived from discourse.   

Based on these points, I recommend to revise this paper focusing on what can we learn  from curent policies/programs for resilient urbanization and how to apply them to developing countries or Parkistan.